# Collaborative Coding with Mixed Initiative LLMs

## Abstract

Large Language Models (LLMs) is quickly arising as a partner for users to solve complex task through multiple interaction turns. To study such interaction, we introduce CoWebDesign, a task and evaluation framework for multi-turn User-LLM interaction for collaborative coding task, where a user work with an LLM assistant to design a website. Most existing LLM assistant work study single-initiative settings, where the LLM assistant generates only output attempts or only clarifying questions to ask the user. We demonstrate both are suboptimal: attempting to predict at every turn is inefficient, as it significantly increases interaction length. Asking questions at every turn is ineffective, as LLM are not very good at asking good clarifying questions consecutively without attempting the task. Given these tradeoffs, we propose mixed-initiative interactions, where LLM alternates between generate clarifying questions and attempting an output, achieving 99% of the output quality from such single-initiative interactions with conversations that are only 55% as long. Lastly, we investigate why mixed-initiative interactions are so effective, demonstrating that mixed-initiative interactions can lead to more helpful user answers to clarifying questions and more efficient communication between the user and assistant.

## 1 Introduction

Collaborative programming is a fundamentally interactive process; however, the current experience of programming with LLM assistants is often more iterative than interactive. Initial user queries for complex coding tasks are typically incomplete and underspecified, yet it is the norm for LLMs to generate a full output attempts at each interaction turn. While this interaction pattern can lead to high quality outputs, it is also inefficient for both Users and LLM assistants. For users, interpreting, evaluating, and constructing freeform feedback to correct the LLM's code outputs imposes a high cognitive load. For LLMs, generating intermediate output attempts leads to long context lengths and high inference costs.

To address these issues with current User-LLM interactions, we study *mixed-initiative* interaction patterns with LLM-assistants for collaborative tasks. Here, over the course of a conversation the LLM may proactively elicit feedback from users by asking them questions instead of always presenting their best guess output, reducing the above costs to the user and LLM.

While prior work (Zhang et al., 2025; Li et al., 2024a) has explored training and evaluating LLMs for multi-turn interactions, they primarily study *single-initiative* interactions. In other words, interactions are furthered by only the user, by providing feedback to the LLM's previous output prediction, or only the LLM, by asking clarifying questions at each turn before finally predicting on output. Recent work (Wu et al., 2025; Laban et al., 2025) address settings that allow for *mixed-initiative* interactions, where conversations may be furthered by either party, but do not explicitly track the LLM's actions or consider the subsequent cost to the user or assistant in their evaluations. Evaluating such costs is critical as users task LLMs with increasingly complex tasks where freeform feedback is more difficult to produce and as LLM predictions become increasingly expensive to generate (Snell et al., 2024; Muennighoff et al., 2025).

We address these challenges by developing evaluations for multi-turn interactions in a collaborative web design task, CoWebDesign: given a user's description of webpage, the user and LLM must work together to generate the code to generate a webpage. In our setting, each system is evaluated

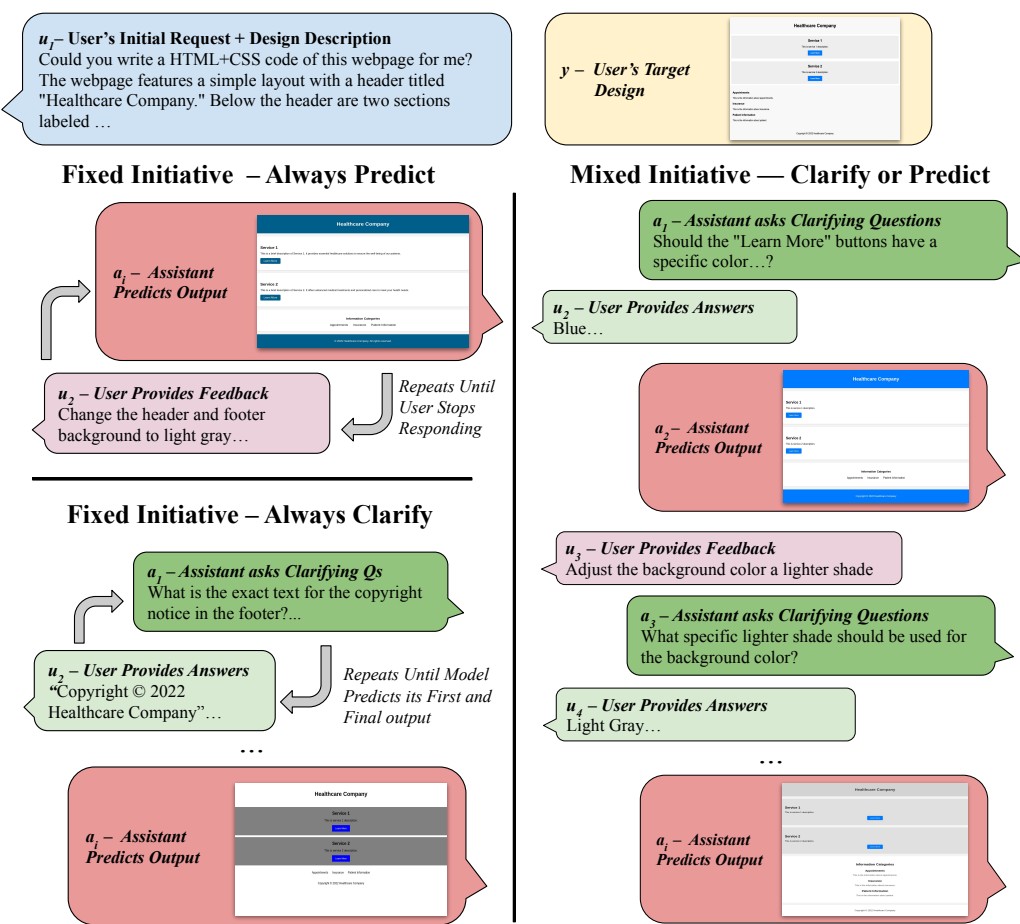

Figure 1: On the top, we present COWEBDESIGN example. In the bottom, we present how users interact with LLMs starting from their initial request $u_1$ containing a textual description of their target design $y$. Over multiple turns of interaction, LLMs must understand and implement the webpage to match the target design. On the left, we depict the two most common methods for designing benchmarks for multi-turn interaction, which are based on the assistant always predicting output attempts or always asking the user clarifying questions until predicting a final output. On the right, we depict mixed initiative interaction patterns explored in this work, where assistants have the ability to utilize multiple strategies through the interaction.

based on its *efficacy*, how closely does the system's final output compares to the user's desired webpage design, and *efficiency*, what is the load imposed on the user and the computational cost on the assistant. While we focus on this task of collaborative web design, our framework for evaluating multi-turn interactions is general and can be applied to many collaborative tasks.

We compare the performance of LLM coding assistants over different single-initiative and mixed-initiative interaction patterns with two LLMs as coding agents (Dubey et al., 2024; Team, 2025b). In both cases, we find mixed-initiative strategies are more effective than single-initiative interactions in the tradeoff between interaction efficacy and efficiency. We find that simple mixed-initiative strategies such as alternating between LM-assistant turns of asking clarifying questions and attempting output predictions achieves 99% of the performance of the standard single-initiative interactions while only requiring interactions to be 55% as long.

We then take a look at why mixed-initiative interactions are able to achieve more favorable tradeoffs between interaction length and output quality. Through ablations and analysis, we identify two key factors to the success of such strategies. First, we find that intermediate output generations in

Table 1: Comparison between the setting proposed in our work, COWEBDESIGN, with settings from prior work for benchmarking multi-turn coding for LLM assistants: CollabLLM, ColBench, and Sketch2Code. While the settings in CollabLLM and ColBench can support mixed-initiative interactions, they do not distinguish differing actions from the LLM assistant nor quantify the differing load imposed on the user based on the LLM's actions (i.e., answering questions or evaluating and providing freeform feedback to an LLM's output prediction). For evaluation, many works utilize automatic metrics such as BLUE or pixel-level overlap between visual elements.

| | Multimodal | Mixed-Initiative | Evaluations | Open-source LLMs |
|---|---|---|---|---|
| Sketch2Code (Li et al., 2024a) | Yes | No | Auto + CLIP | No[1] |
| CollabLLM (Wu et al., 2025) | No | Untracked | Auto | Yes |
| ColBench (Zhou et al., 2025) | Yes | Untracked | Auto + CLIP | Yes |
| COWEBDESIGN(Ours) | Yes | Tracked | VLM-Judge | Yes |

mixed-initiative interactions allow the models to make assumptions about an underspecified task and communicate them to the user, who may then identify which assumptions were incorrect and correct them via feedback. Second, we demonstrate that intermediate generations also broadens the scope of possible clarifying questions and allows users to provide more informative answers by contextualizing them in the model's prior attempts.

## 2 TASK: HUMAN-LLM COLLABORATIVE WEB DESIGN

Our proposed task, COWEBDESIGN, is based on the realistic scenario where users must collaborate with an LLM assistant to implement a website they have designed. We center our study of multi-turn interactions around this scenario for several key reasons. First, such coding and media generation tasks represent a significant proportion of real user requests to LLM assistants (Zhao et al., 2024; Chatterji et al., 2025). Second, this task is its suitability for novel user-simulation settings that do not rely on prompting users with the full, disambiguated input task (Andukuri et al., 2024; Pan et al., 2025) nor the gold model output (Zhang et al., 2025), which we further discuss in Section 2.3.

Lastly, the task provides testbed for collaborative interactions where non-expert users can easily provide feedback. In collaborative web design, simulated users can easily evaluate and critique with attempted model outputs. In contrast, prior works studying multi-turn interactions are primarily designed around on settings where users cannot reliably evaluate the correctness of an LLMs predicted outputs, like solving math problems (Li et al., 2025; Laban et al., 2025), providing clinical advice (Li et al., 2024b), or general question answering (Zhang et al., 2025; Chen et al., 2025). Because users cannot reliably identify incorrect model predictions, they incentive systems that only ask clarifying questions before predicting a single, final output and are poorly suited for the types of collaborative interaction patterns explored in this work. While several other recently proposed benchmarks have also studied multi-turn settings involving coding with an LLM assistant, man benchmarks are often designed around interaction frameworks based on the needs of these other tasks. We provide a breakdown of such multi-turn coding benchmarks in Table 1.

### 2.1 TASK DEFINITION

Each interaction is based around a gold image $y$ of the website design the user wants the LLM's help to implement. To achieve this, users present an initial input query $u_1$ containing a brief textual description of their desired website design $y$. Assistants then produce their response $a_1$, and users and assistants then continue to take turns $(u_2, a_2, ...)$ until the assistant produces its final code output.

At each turn, the assistant can advance the conversation $\psi(a_i) = $ Clarify by asking a clarifying questions to the user, or on the user $\psi(a_i) = $ Predict by predicting its best-guess output and awaiting user's freeform feedback. At each turn, the assistant's decision in its response $a_i$ to Clarify or Predict an output attempt determines whether the user's subsequent response $u_{i+1}$ answers the assistant's clarifying questions or critiques their predicted output attempt.

While prior works (Li et al., 2024a;b) have explored similar settings where models must decide to Clarify or Predict an output attempt at each turn, these works have exclusively studied *single-*

*initiative* interaction patterns, where the LM assistant always places the initiative on exclusively the user or the assistant ($\psi(a_1) = \psi(a_2) = \dots$) in this work we examine the benefits of *mixed-initiative* interaction settings where the initiative can go back and forth between the user and assistant (e.g., $[\psi(a_1) = \texttt{Clarify}, \psi(a_2) = \texttt{Predict}, \dots]$). This setting only considers conversations that are oriented toward a single goal, and does not cover situations where the user's goals may change over the course of the conversation (Zhao et al., 2024; Chen et al., 2025).

**Dataset Implementation** We source example website designs $y$ from the WebSight dataset (Laurençon et al., 2024), a dataset containing 2 million synthetically generated webpage screenshots. All website designs in this dataset are implemented in plain HTML+CSS. To initiate interactions based on these images $y$, we generate initial user requests $u_i$ by prompting GPT-4o to construct concise descriptions of each webpage design $y$ (prompt in Appendix B). Using this, we construct a test set of 500 examples which we use throughout this work for all experiments.

## 2.2 EVALUATIONS

We evaluate each interaction along two axes: *efficacy* and *efficiency*.

**Efficacy Metrics** We evaluate the efficacy by rating the quality of the LLM assistant's final output prediction ($a_i$ where $\phi(a_i) = \texttt{Predict}$). To do this, we first render the model's final output prediction to produce a screenshot $\hat{y}$ and directly compare it against the original, gold design image $y$ using VLM-as-a-Judge (Zheng et al., 2023; Joseph et al., 2025). Our judge system takes both images as input and is tasked with rating the quality of the replication on a 1-10 scale: $(y, \hat{y}) \to [1, 10]$.

We experiment with both API-based (GPT-4o, GPT-4o-mini (Achiam et al., 2023)) and open-sourced options (Qwen-2.5-VL (Team, 2025a)) for our VLM-as-Judge system. To compare these methods and test their veracity, the authors of this paper manually annotate a set of 100 pairwise comparisons, labeling the screenshots from two replications of the same target webpage and identifying which is better, or if they are tied. We then evaluate VLM-as-Judge systems by scoring each replication, and comparing the numerical scores to determine the model's pairwise judgment.

In Table 2 we report both three-way accuracy and Cohen's kappa of each system's predictions, evaluated against our manually annotated labels. Overall, we find that performance is heavily tied to model size, with almost random agreement with Qwen2.5-VL-7B variants and GPT-4o-Mini, and significant improvements with their larger counterparts, Qwen2.5-VL-32B and GPT-4o. We see the best performance when using GPT-4o with fair agreement to human labels, and thus use this system as our evaluation metric throughout the remainder of this work. We include details on prompts and annotation methods in Appendix B. In the appendix, we

Table 2: Comparing automatic efficacy evaluation models (Pointwise VLM-as-Judge) against human annotation ($n = 100$) using Cohen's Kappa and three-way accuracy.

| Model | Cohen's $\kappa$ | Accuracy |
|---|---|---|
| GPT-4o | 0.31 | 54 |
| GPT-4o-mini | 0.09 | 40 |
| Qwen2.5-VL-7B | 0.04 | 34 |
| Qwen2.5-VL-32B | 0.18 | 46 |

also explore alternative pairwise VLM-as-Judge variants, which demonstrate stronger human agreement than pointwise methods presented above. However, we do not use such systems for evaluations in this work due to computational and cost constraints.

**Efficiency Metrics** When evaluating an interaction, we weigh the efficacy of the interaction (i.e., the quality of the final model output) against the efficiency or cost of the interaction. We consider two metrics: the length of the interaction in turns and the length of the interaction as the total number of assistant and user characters.

Prior work (Li et al., 2024a) has demonstrated that users prefer answering clarifying questions over provided freeform feedback responses, thus it may be advantageous to rate efficiency by tracking the number of each user response type in the interaction. Wu et al. (2025) uses a LLM-as-a-Judge to evaluate "interactivity" of conversation, aiming to judge how engaging users would find the interactions.

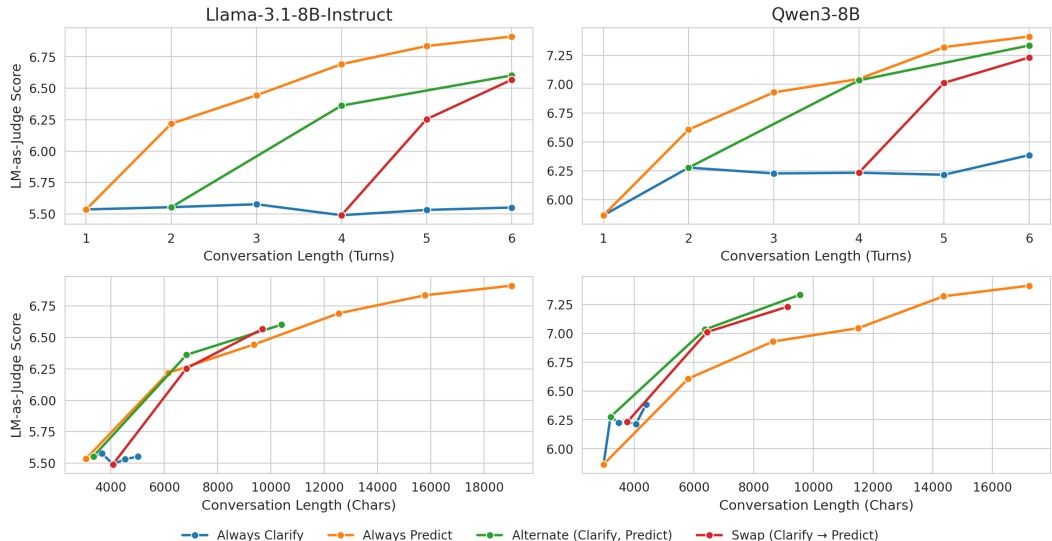

Figure 2: The model output quality as the interaction progresses for each interaction pattern. On the left, we plot Llama-3.1-8B-Instruct model, on the right, we plot Qwen3-8B model as coding agent. The plots on the top row represent performance versus the length of the interaction in turns, while the bottom plots represent performance versus length of the conversation in characters. As we discuss in Section 2.2, the latter is a more apt representation performance versus the computational cost imposed on the model and load imposed on the user.

## 2.3 USER SIMULATION

Simulating users has become a prominent method for evaluating LLM assistants across a range of multi-turn settings such as question-answering (Chen et al., 2025), writing (Andukuri et al., 2024), and coding (Wu et al., 2025). Typically, simulating user responses is done by prompting LLMs with either a fully-specified version of the input request (Wu et al., 2025; Laban et al., 2025), or by conditioning on the gold output (Zhang et al., 2025). Such methods for user-simulation, however, are prone to pitfalls such as generating over-informative answers to or leaking gold target outputs (Lin & Tomlin), and can allow LLMs to solve tasks over a single round of interaction by asking the user open-ended questions (i.e., "Can you tell me more?").

In this work, we specify goals to our simulated user by conditioning on the reference design image $y$ rather than the target output of the LLM system (i.e. the gold code implementation of the reference design). This allows us to specify the user's goals while preventing simulated users from inadvertently leaking the task or solution to the model. In the case where the model's most recent utterance $a_i$ was to predict an output attempt $\phi(a_i) = \texttt{Predict}$, we render the model's prediction to get a screenshot of the model's prediction $\hat{y_i}$ and task the user with generating freeform feedback $(y, \hat{y_i}) \rightarrow u_{i+1}$. In the case where the model asked clarifying questions $\phi(a_i) = \texttt{Clarify}$, we task the user with answering the the model's questions based on the gold target image $y$ and the screenshot from the system's most recent prediction $\hat{y_j}$, if the model has made one $(y, a, \hat{y_j}) \rightarrow u_{i+1}$. Throughout all our experiments, we use GPT-4o as our user simulator system, and provide our full prompts in Appendix B.

## 3 EXPERIMENTS

We evaluate models over range of single and mixed-initiative interaction patterns described below. For each interaction pattern we experiment we report results for all interaction length up to a maximum length of 6 ($M \in \lceil 1, 6 \rceil$). We repeat all experiments with two different LLM base models: Llama-3-8B-Instruct (Dubey et al., 2024) and Qwen3-8B (Team, 2025b). Below, we outline the different interaction patterns experimented with in this work.

## 3.1 COMPARED INTERACTION METHODS

**Single-Initiative Baselines**    Most prior work establishing benchmarks for multi-turn interactions focus on one of two single-initiative settings:

- **Always Predict** reflects the most common interaction mode of LLMs where each assistant turn $a_i$ contains an attempted output prediction and does not probe the user for any additional information regarding the task.
- **Always Clarify** represents an alternative approach, where for all turns up until the final assistant turn ($i \in [1, M-1]$) the assistant asks the user clarifying questions. The system then, for its final turn $a_M$, predicts an output.

**Mixed-Initiative Methods**    We present two variants of mixed-initiative trajectories:

- **Alternate (Clarify, Predict)** here, we consider the most basic mixed-initiative method of alternating between between asking the user clarifying questions (turns $i \in \{1, 3, 5\}$) and predicting an outputs (turns $i \in \{2, 4, 6\}$).
- **Swap (Clarify → Predict)** This strategy is grounded in the intuition that clarifying questions may be helpful in the early stages of a conversation for getting clarification regarding to high-level details about the website design, but less helpful making more fine-grained adjustments to the implementation. Here, for turns $i \in [0, M/2]$, the assistant turn $a_i$ contains a clarifying question. For the remaining turns $i \in [0, M/2]$ swaps its strategy by always predicting an output and elicits freeform feedback from the user.

## 3.2 RESULTS

We report our main results in Table 2. Looking first at the our two single-initiative interaction patterns, we find that Always-Predict shows consistent improvements in output quality after each interaction turn for both LLM systems, with diminishing returns in later turns of the interaction. In contrast, the Always-Clarify method demonstrates mixed or negligible changes after subsequent turns of the interaction, with notable exception in a small improvement in Qwen3-8B output quality after asking a single round of clarifying questions.

When comparing the results of the Always-Predict interactions against our two mixed-initiative methods, we see that its performance falls short when compared across conversations of equal turn length. This pattern is reflective of the interpretation of Always-Predict interactions as an upper bound for output quality at each turn at the cost of maximizing computational cost and effort from the user. When comparing these methods on performance versus conversation length in characters, however, we see that mixed-initiative methods are able to flip this trend, ultimately achieving stronger eficacy versus efficiency tradeoffs, particularly for the Qwen3.

While these results above suggest that clarifying questions asked by the systems are not helpful for learning more about the user's goal's, looking at our mixed-initiative interaction settings tell a different story. In particular, looking the results from our Swap (Clarify → Predict) experiments, we see that the prior rounds of clarifying questions actually increase the rate of improvement from rounds of freeform feedback when compared against the early rounds of our Always Predict baseline. The benefits of these early rounds of question asking are further reinforced when looking at the the tradeoff between output quality and conversation length when measured in number of characters.

# 4 ANALYSIS

Interactive Human-LLM collaboration is a recent research topic, and very little work provides analysis on when and where model fails. We present analysis breaking down key components for successful human-LLM collaboration.

We analyze where the gains in mixed-initiative interaction may be coming from. In particular, we look at two possible sources of the gains: (1) the model's ability to generate intermediate solution attempts during a conversation (Section 4.1) and (2) enhancing the scope of useful questions that can be asked when the user is given access to the model's current attempt (Section 4.2).

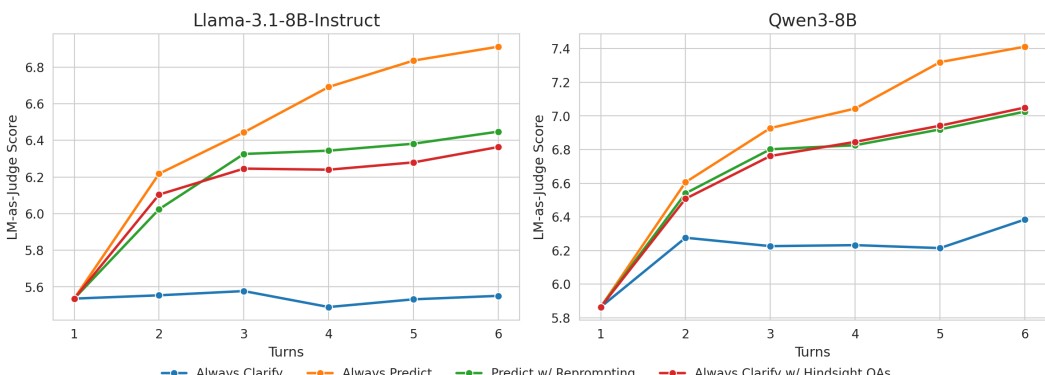

Figure 3: Model output quality results after ablating intermediate model output attempts from the context. Predict w/ Reprompting simply removes the intermediate output attempts from the context of from our Always Clarify baseline's interactions. Always Clarify w/ Hindsight QAs further paraphrases these conversations into ones in the style of our Always Clarify baseline, where the assistant asks and the user answers clarifying questions at each conversation turn.

### 4.1 UNDERSTANDING THE ROLE OF INTERMEDIATE OUTPUT GENERATIONS IN INTERACTIONS

A significant distinction between the Always-Clarify method and the other interaction patterns is the absence of intermediate output generations. To investigate their role, we compare the quality of LLM outputs after removing them from interactions in the following two settings:

- **Always Predict w/o Intermediate Output**: LLM directly generate an output in a single conversation turn after appending all user-provided feedback from our Always-Predict baseline. Concretely, using the Always-Predict interaction $u_0, a_0, \ldots, u_i, a_i$, we concatenate all user-turns to the user's initial instruction $u_0^{\text{Reprompt}} = [u_0, u_1, \ldots, u_i]$ and task the model with predicting an output $u_0^{\text{Reprompt}}$.

- **Always Clarify w/ Hindsight QAs**: We change the input format from free-form feedback to clarifying question answer pairs from the above method. To accomplish this, we prompt GPT-4o to paraphrase each turn of user feedback into QA pairs, $u_i \rightarrow (a_{i-1}^{\text{Hindsight}}, u_i^{\text{Hindsight}})$. We then prompt the LM with the full interaction $(u_0, a_1^{\text{Hindsight}}, u_1^{\text{Hindsight}} \ldots u_i^{\text{Hindsight}})$ and task the LLM with predicting an output webpage on the subsequent turn.

We report the results of these ablations in Figure 3. Here, we find that both our ablation methods perform similarly, significantly outperforming our Always Clarify baseline while lagging behind the performance of our Always Predict setting. This demonstrates two things. First, that the poor performance of Always-Clarify interactions is not due to the LM's inability to incorporate additional instruction specifications when presented in Clarifying Question-Answer format. Second, that intermediate generations are important to model performance.

Figure 4 depicts an example demonstrating the communicative role that intermediate generations play. Here, we see that our Always-Predict initial prediction was able to correctly assume the user's desired intent in their descriptions of the navigation bar and background color based on only the user's initial description. Thus, there was no mention of either of these two features in any of the later rounds of feedback. In contrast, the Reprompting ablation makes two different assumptions regarding the shade of white of the background and formatting of the navigation bar. This highlights how intermediate generations can actually allow users to avoid over-specifying their instructions, relying on the assistant to make reasonable assumptions based on the given information. Later turns of the interaction, thus, can be focused on correcting only details where the model's assumptions were incorrect.

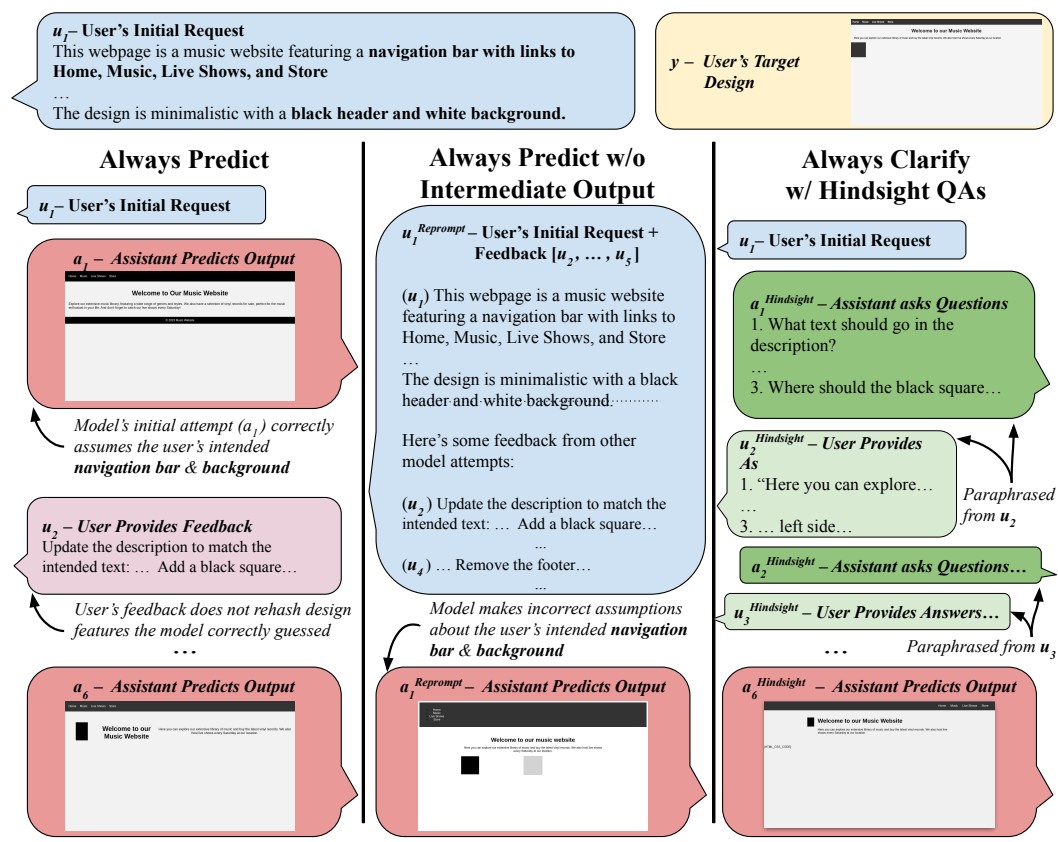

Figure 4: Example comparing an Always Predict interaction against two others where intermediate output attempts have been removed, while the user-provided details and specifications regarding their target design are maintained. In the Always Predict interaction, the system made correct assumptions with respect to the background color (light gray / off-white shade) and navigation bar format (items listed horizontally, left justified). This intermediate prediction allows the user to forgo any further details regarding either of these elements in later rounds of the conversation. When intermediate predictions are removed, we see that systems may change their assumptions regarding the navigation bar (vertically listed, right justified) or background elements (true white shade).

## 4.2 MIXED-INITIATIVE INTERACTIONS EMPOWER QUESTION ASKING SYSTEMS

Here, we examine whether Mixed-Initiative interactions can elicit more helpful answers to clarifying questions from the user. We hypothesize that giving the user access to intermediate output generations can help users to provide better answers to LLMs' clarifying questions.

To evaluate this, we construct conversation histories from the interactions from our Always-Predict baseline $(u_1, a_1, ...a_i)$. We then replace the last assistant turn with QA pairs that were hindsight generated from the subsequent turn of user feedback, following the same process as described in Section 4.1, giving us $(u_1, a_1, ...a_i^{hindsight})$.

Table 3: Comparing user simulation w/ and w/o image.

| Model | % of data | | |
| --- | --- | --- | --- |
| | w/ img is Better | Tied | w/o img is Better |
| Llama | 46 | 28 | 26 |
| Qwen | 46 | 30 | 24 |

We then generate answers to the hindsight-generated clarifying questions $a_i^{hindsight}$ using two different user-simulators. The first is our standard setting where users are given access to the previous model output generation. In the second, we task the users to answer questions based on only their gold target $y$. We then compare the model's subsequent output generations after observing the an-

swers from each of these two user simulators, **W/ Img** and **W/o Img** using our GPT-4o pairwise VLM-as-Judge.

Table 3 presents the results. In both models, the answers from our user simulator with access to the prior image are significantly more helpful to the assistant than answers from the user simulator without access.

## 5  RELATED WORK

**Multi-Turn Benchmarks**  Numerous prior works have established benchmarks for multi-turn interaction. Closely related to our work, Li et al. (2024a) proposed Sketch2Code benchmark for interactive web-design; however, they study generating webpages from wire frame sketches, which lies beyond the capability of open-sourced LLMs. We study more typical user scenario, where the task description is given in text rathre than sketch. Likewise, works have established multi-turn benchmarks for LLM-agents (Yao et al., 2024; Barres et al., 2025), factoid QA (Zhang et al., 2025), and Medical QA (Li et al., 2024b). The tasks explored in these benchmarks, however, are less suitable for collaborative interactions, as users may not be able to provide feedback to erroneous intermediate output predictions to factoid or medical questions. In contrast, our collaborative setting does not have the same risks associated with presenting intermediate, imperfect solutions to the user.

**Code Generation**  Numerous works have established benchmarks for generic code generation (Jimenez et al., 2023; Li et al., 2024a). Recent work has built upon such benchmarks, testing the ability for models to recognize and resolve ambiguity in such benchmarks with the use of clarifying questions (Vijayvargiya et al., 2025). Zhou et al. (2025) proposes similar coding benchmarks for front-end and back-end coding tasks as well as learning algorithms for training coding agents in interactive settings. Recent work has also explored visual settings similar to web design, like generating code for multimodal domains (Yang et al., 2024) and generating slide decks from instructions (Ge et al., 2025).

**LLMs and Ambiguity**  Prior work has explored identifying ambiguities in user requests, which may be used to determine whether to ask a clarifying question or to predict an output, as an uncertainty estimation task (Cole et al., 2023; Zhang & Choi, 2023). Such techniques, however, are not directly aligned with our goals in mixed-initative interaction where inputs are always ambiguous, even after multiple rounds of clarification, and systems must weigh the information gained from an action against its cost. Other works have studied the prevalence of ambiguity, and whether LMs can identify then, in a range of other tasks not explored in this work, such as NLI (Liu et al., 2023), co-reference resolution (Yuan et al., 2023), and translation (Voita et al., 2019). In such settings, however, ambiguities can typically be resolved within a single turn of interaction, compared to the webdesign task in this work where LLMs are able to continually improve their output predictions over many rounds of interaction.

## 6  CONCLUSION

We study mixed-initiative, multi-turn interactions with LLM assistants in a collaborative coding task, COWEBDESIGN. In this task, a simulated user must work together with an LLM assistant to implement their intended website design by communicating and specifying their design over multiple interaction turns. While most multi-turn interaction benchmarks are designed for single-initiative interactions, we find that mixed-initiative interactions are able to achieve strong efficiency versus efficacy tradeoffs. Specifically, we find that giving the LLM the option to either generate clarifying questions or an output attempt allows them to to achieve 99% of the output quality from single-initiative interactions with conversations that are only 55% as long. We further investigate why mixed-initiative interactions are so effective, demonstrating that mixed-initiative interactions can lead to more helpful user answers to clarifying questions and more efficient communication between the user and assistant.

## 7 REPRODUCIBILITY STATEMENT

All results and artifacts from this work are from open-source or publicly available models. We include all necessary prompts for reproducing the datasets and settings in this work in our Appendix. Additionally, we plan to release code for all experiments and data generation upon acceptance.

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

Table 4: Comparing automatic efficacy evaluation models (Pointwise VLM-as-Judge) against human annotation ($n = 100$) using Cohen's Kappa and three-way accuracy.

| Model | Method | Cohen's $\kappa$ | Accuracy |
|---|---|---|---|
| GPT-4o | Pointwise | 0.31 | 54 |
| | Pairwise | 0.70 | 81 |
| GPT-4o-mini | Pointwise | 0.09 | 40 |
| | Pairwise | 0.32 | 54 |
| Qwen2.5-VL-7B | Pointwise | 0.04 | 34 |
| | Pairwise | 0.23 | 50 |
| Qwen2.5-VL-32B | Pointwise | 0.18 | 46 |
| | Pairwise | 0.49 | 68 |

## A  EVALUATION DETAILS

Our manually labeled set of 100 examples is performed by randomly sampling 100 interactions from LLama-3.1-8b-Instruct using the Always Predict interaction pattern. Of these 100 interactions, we randomly sample two model predictions and label which rendered screenshot more closely matches the gold target design.

In addition the the pointwise evaluation metrics used in this work, we additionally explore pairwise evaluation methods. For our pairwise methods, we prompt models compare two replications to determine which is better or if they are tied: $(y, \hat{y}_1, \hat{y}_2) \rightarrow \{\hat{y}_1, \hat{y}_2, \texttt{TIE}\}$. In Table 4, we report our results and find that pairwise methods, overall, have substantially higher agreement with human labels. Due to the increased cost of running such evaluations, however, we rely on the pointwise judge method throught our main experiments.

### A.1  PROMPT DETAILS

We include all prompts used for data generation, user simulation, and VLM-as-Judge evaluation. We additionally include the system prompt for instructing our evaluated coding assistant LLMs.

## B  IMPLEMENTATION DETAILS

---

**Instruction Generation Prompt**

Generate a concise, two to three sentence description of the this webpage screenshot and its layout.

---

Figure 5: Prompt for generating website descriptions for initial user requests $u_1$

648
649
650
651
652
653
654
655
656
657
658
659
660
661
662
663
664
665
666
667
668
669
670
671
672
673
674
675
676
677
678
679
680
681
682
683
684
685
686
687
688
689
690
691
692
693
694
695
696
697
698
699
700
701

**Coding Assistant System Prompt**

# Task
You are web developer assistant who specializes in HTML and CSS.
Users come to you with a description of a website they've designed that they want your help to implement.
Your task is to have a conversation with the user to understand their design and create an implementation of it.

# Instructions
During each turn of the conversation, you may respond with either an implementation of their design or by asking the user three clarifying questions.
All implementations should be a single, self-contained HTML file that uses HTML and CSS to produce a webpage that strictly follows the user's description.
Include all CSS code in the HTML file itself.
Do not hallucinate any dependencies to external files.
All clarifying questions should help you understand the user's exact design specifications.
Pay attention to things like size and position of all the elements, as well as the overall layout.
You may assume that the page is static and ignore any user interactivity.

# Formatting
Each response should immediately begin with "[[CODE]]" if you decide to generate an implementation of their design or "[[CLARIFY]]" if you decide to ask the user clarifying questions.
Your implementation or clarifying questions should immediately follow, starting on a new line.
Do not include any additional text.

If you decide to generate an implementation of their design, your response should look like this:
[[CODE]]
{{HTML_CSS_CODE}}

If you decide to ask the user clarifying questions you should respond like this:
1: {{FIRST_QUESTION}}
2: {{SECOND_QUESTION}}
3: {{THIRD_QUESTION}}

Figure 6: System Prompt for all LLM coding assistant systems

**User Simulation Freeform Feedback Prompt**

# Design Review Instructions

You are a design reviewer helping a code agent implement an HTML webpage. You will receive two images:
1. **Target design** - the intended webpage layout
2. **Current implementation** - what the code agent has built so far

## Your Task
Compare the current implementation against the target design and provide feedback in the form of three specific, actionable instructions to help the code agent improve their work.

## Critical Guidelines
- Your feedback **MUST** be **strictly** based on the provided screenshots.
- You should **NEVER** make things up.
- The agent is not supposed to know about the target design, so you should **NEVER** mention the target design in your response, nor should you ever give out any HTML content to the agent.
- Only mention what needs to change, not what's already correct

## Formatting
You may compare and analyze the two webpages step by step.
Once you are ready, provide your final feeback on a new line using triple quotes like this:
Feedback: """
1: {{FIRST_FEEDBACK_INSTRUCTION}}
2: {{SECOND_FEEDBACK_INSTRUCTION}}
3: {{THIRD_FEEDBACK_INSTRUCTION}}
"""

# Inputs
Target design:

Current Implementation:

Figure 7: Prompt for simulating freeform user feedback to model predictions

**User Simulation Prompt for Answering Clarifying Questions (w/o img)**

# Design Review Instructions

You are a design reviewer helping a code agent implement an HTML webpage. You will receive one images:
1. **Target design** - the intended webpage layout

You will also recieve a list of one or more questions asked by the code agent.

## Your Task
Answer the code agent's questions based on the target design.
Your answers should help the code agent improve their work.

## Critical Guidelines
- Your answers **MUST** be **strictly** based on the provided screenshots.
- You should **NEVER** make things up or provide any information more than what the agent asks for.
- The agent is not supposed to know about the target design, so you should **NEVER** mention the target design in your response, nor should you ever give out any HTML content to the agent.

## Formatting
Your answers should concise and at most one sentence long.
When possible, answers should be a single word or phrase.
You may consider each question step by step before providing your answers.
Once you are ready, provide your final answers on a new line using triple quotes like this:
Answers: """
1: {{ANSWER_TO_FIRST_AGENT_QUESTION}}
2: {{ANSWER_TO_SECOND_AGENT_QUESTION}}
..."""

# Inputs
Target design:

Agent's questions:

Figure 8: Prompt for simulating user responses to model clarifying questions when there is no prior model prediction attempt in the conversation history.

**User Simulation Prompt for Answering Clarifying Questions (w/ img)**

# Design Review Instructions
You are a design reviewer helping a code agent implement an HTML webpage.
You will receive a screenshot of the target design.
You will also recieve a list of one or more questions asked by the code agent.

## Your Task
Please answer the agent's questions based on the provided target design.

## Critical Guidelines
- Your answers **MUST** be **strictly** based on the provided target design screenshot.
- You should **NEVER** make things up or provide any information more than what the agent asks for.
- The agent is not supposed to know about the target design, so you should **NEVER** mention the target design in your response, nor should you ever give out any HTML content to the agent.

## Formatting
Your answers should concise and at most one sentence long.
When possible, answers should be a single word or phrase.
You may consider each question step by step before providing your answers.
Once you are ready, provide your final answers on a new line using triple quotes like this:
Answers: """
1: {{ANSWER_TO_FIRST_AGENT_QUESTION}}
2: {{ANSWER_TO_SECOND_AGENT_QUESTION}}
..."""

# Inputs
Target design:

Current Implementation:

Agent's questions:

Figure 9: Prompt for simulating user responses to model clarifying questions when there is a prior model prediction attempt in the conversation history.

**Pointwise VLM-as-Judge Prompt**

# Instructions
You are a design reviewer evaluating a code agent's implementation an HTML webpage. You will receive two images:
1. Target design - the intended webpage layout
2. Agent's implementation - what the code agent has built

## Your Task
Your task is to judge the agent's implementation by comparing it against the intended webpage and assigning it a score from 1 (worst) to 10 (best).

## Formatting
You may compare and analyze the two webpages step by step.
Once you are ready, your provide a score on a new line like this:
{{STEP_BY_STEP_COMPARISON}}
Score: {{FINAL_SCORE}}

# Inputs
Target design:

Agent's implementation:

Figure 10: Pointwise VLM-as-Judge prompt for evaluating replicated webpage screenshots $\hat{y}$ against gold design images $y$

**Pairwise VLM-as-Judge Prompt**

# Instructions
You will be provided with screenshots of three webpages.
The first one is the original design.
The next two are attempted replications: "Attempt 1" and "Attempt 2".
Your task is to determine which of the two replications more closely matches the original design, or if they are tied.
You should respond by selecting "1", "2" or "Tie".

## Formatting
You should first compare the two replications step-by-step before providing your final answer.
When you are ready, provide your final response on a new line like this:
{{STEP_BY_STEP_COMPARISON}}
Final Answer: {{1/2/Tie}}

# Inputs
Original Design:

Attempt 1:

Attempt 2:

Figure 11: Pairwise VLM-as-Judge prompt for comparing two replicated webpage screenshots $\hat{y}_a$ and $\hat{y}_b$ against gold design images $y$

