# OpenReview forum: "Collaborative Coding with Mixed Initiative LLMs"
_ICLR.cc/2026/Conference — ICLR 2026 Conference Withdrawn Submission_

### Official Review · Reviewer_BHCs · 2025-10-30

**Soundness:** 3
**Presentation:** 3
**Contribution:** 2
**Rating:** 2
**Confidence:** 4

**Summary:**

The benchmark an important question in human llm collaboration, specifically investigating mixed initiative vs single initiative collaboration between humans and llm. The idea is simple, but framed cleanly and the insights / takeaways are interesting. Overall, I believe this paper has significant potential. However, as is, this paper would be more suited to a different venue that focuses more on analysis of Human-LLM behavior. The content is currently too narrow and the contributions are too limited to be accepted as a completed work.

**Strengths:**

- Benchmark tackles a significant and important question in human llm collaboration.
- The idea is simple, but framed cleanly and the insights/takeawys are interesting

**Weaknesses:**

- Experiments need evaluate on more models (at least all major providers or all open source models).
- The methods in the paper are currently relatively simple. In fact, I would consider all approaches currently in the paper as a baseline method. There needs to be a method that actually improves human-llm collaboration.
- The "mixed initiative" method underperforms in terms of LLM as a judge score, with the gains coming mostly from efficiency / turns.

**Questions:**

My main questions is why there are so little models and methods. It feels that the experiments described in the paper would not be prohibitively expensive to run.

---

### Official Review · Reviewer_Aaos · 2025-10-31

**Soundness:** 3
**Presentation:** 3
**Contribution:** 2
**Rating:** 4
**Confidence:** 4

**Summary:**

The paper introduce CoWebDesign, a task and evaluation framework for multi-turn UserLLM interaction for collaborative coding task. The user and AI assistant together design a webpage in HTML + CSS. To evaluate the quality of the web design, the model use VLM judge to compare the rendered webpage against the gold webpage image to score in 1 to 10 scale. The model evaluate 2 LLMs as the AI assistants: Llama 3 8B  and Owen3 8B.

**Strengths:**

The paper is easy to read, the task is interesting

**Weaknesses:**

The evaluator performance is very low with the best evaluator GPT-4o achieving only 54 accuracy
There is no evaluation on the user simulator
Only evaluate 2 LLMs as the AI assistant

**Questions:**

See weaknesses

---

### Official Review · Reviewer_cqwv · 2025-10-31

**Soundness:** 2
**Presentation:** 2
**Contribution:** 2
**Rating:** 4
**Confidence:** 4

**Summary:**

This paper studies multi-turn interaction patterns for collaborative web design. The authors introduce CoWebDesign, where users work with LLM assistants to implement websites. They compare single-initiative approaches (always predicting or always asking questions) against mixed-initiative patterns that alternate between both. Results suggest mixed strategies achieve comparable quality with shorter conversations. Ablations indicate intermediate outputs help by allowing implicit communication and enabling more informative user responses.

**Strengths:**

1. The web design task enables user simulation based on target images rather than gold code (avoiding information leakage). Critically, the framework tracks different interaction costs—answering questions versus evaluating outputs and providing feedback—which prior work ignored.
2. Results hold across two models. The ablations in Section 4 go beyond performance comparison, showing how intermediate generations enable implicit assumption communication.

**Weaknesses:**

1. Section 3.2 states "We report our main results in Table 2" but no such table exists—presumably referring to Figure 2 instead. More problematic, the paper does not provide numerical tables with actual performance values. Figure 2 shows only line graphs. Without clear numerical data and statistical testing (especially given κ=0.31 evaluation reliability), it's hard to verify the quantitative claims. The main experimental results should be presented in a readable numerical table.
2. The paper frames the work around LLMs deciding when to ask questions versus predict. However, Alternate and Swap follow fixed schedules. The model never makes context-dependent decisions about whether it has enough information.
3. Always Clarify baseline gets no refinement (predict once at end) while Always Predict gets M refinements—seems intentionally weak but doesn't test questions+iteration.
4. Missing details like sampling parameters and the rationale for maximum turns.

**Questions:**

See weaknesses

---

### Official Review · Reviewer_Wufb · 2025-11-11

**Soundness:** 2
**Presentation:** 3
**Contribution:** 2
**Rating:** 2
**Confidence:** 4

**Summary:**

The work studies how to improve LLM's performance in code design interactive environments, where they explore mixed initiative interaction patterns, where assistants have the ability to utilizemultiple strategies through the interaction. The work has reasoned that both extreme cases where the assistant keeps asking questions and providing answers are not ideal. The mix-initiative pattern has improve the performance and scale up well with the context size.

**Strengths:**

- The presentation is clear and the method is easy to understand.
- The problem of studying interactive settings is increasingly important.

**Weaknesses:**

- There are a few false statements to my knowledge, for example:
   ```
   Recent work (Wu et al., 2025; Laban et al., 2025) address settings that allow for mixed-initiative interactions, where conversations may be furthered by either party, but do not explicitly track the LLM’s actions or consider the subsequent cost to the user or assistant in their evaluations.
   ```

    This is not precise/true, for example, in CollabLLM,  they have considered the subsequent cost to the user as far as I understand.

   Another example,
   ```
   While prior works (Li et al., 2024a;b) have explored similar settings where models must decide to Clarify or Predict an output attempt at each turn, these works have exclusively studied single-initiative interaction patterns, where the LM assistant always places the initiative on exclusively theuser or the assistant (ψ(a1) = ψ(a2) = . . . ) in this work we examine the benefits of mixed-initiative interaction settings
   ```

   I don't think this is true, a lots of work that study llms in multiturn settings do not explicitly define these two actions (as also listed by the author in Tab1), meaning that there's already a mixture of both or sometimes, a response can contain both the prediction and clarification in CollabLLM. This seem to view previous works in a very narrow way and the claim that the previous work doesn't track actions may not be a limitation or advatange of this work compared to them. Because we don't know if the optimal interation is coming from a discrete selection of actions or an uncontrained paradigm. I think the setting needs to be discussed more comprehensively.

- Given last two points, it seems the work's novelty comes from an explicit definition of actions where LLM alternates between generating clarifying questions and attempting an output, which poses contraints towards LLM's actions, and this may not be necessary or flexible in some cases.

- In Figure 1, the phenomeno seems to be conditional on the llm type as well. I agree that for small llms, especially those open-sourced ones, the models can be asking questions over and over, but i don't think this is a problem for bigger llms with slightly better context understanding and instruction following. For example, the user can just say, please answer directly and the model stops asking questions in the real world.

- Web design is an interesting task, but i am wondering what prevents this work from applying to other datasets. The other tasks doesn't seem to be more complex than web design and are easier to evaluate as well.


Minor:
- typos:
Line 21:
```
between generate -> between generating
```
line 125:
```
broadens -> boarden
allows -> allow
```

**Questions:**

- What is the key constraint and limitation of your method by defining the explicit actions?
- What prevents the work from applying to other multiturn datasets?
- What is the novelty of this work compared to the existing ones?

---

### Note · Authors · 2025-12-01

I have read and agree with the venue's withdrawal policy on behalf of myself and my co-authors.